# Boosting the Potential of Large Language Models with an Intelligent Information Assistant

**Yujia Zhou**\*
Tsinghua University
zhouyujia@mail.tsinghua.edu.cn

**Zheng Liu**\*
The Hong Kong Polytechnic University
zhengliu1026@gmail.com

**Zhicheng Dou**
Renmin University of China
dou@ruc.edu.cn

## Abstract

The emergence of Large Language Models (LLMs) has significantly advanced natural language processing, but these models often generate factually incorrect information, known as "hallucination". Initial retrieval-augmented generation (RAG) methods like the "Retrieve-Read" framework was inadequate for complex reasoning tasks. Subsequent prompt-based RAG strategies and Supervised Fine-Tuning (SFT) methods improved performance but required frequent retraining and risked altering foundational LLM capabilities. To cope with these challenges, we propose Assistant-based Retrieval-Augmented Generation (AssISTRAG), integrating an intelligent information assistant within LLMs. This assistant manages memory and knowledge through tool usage, action execution, memory building, and plan specification. Using a two-phase training approach—Curriculum Assistant Learning and Reinforced Preference Optimization—AssISTRAG enhances information retrieval and decision-making. Experiments show AssISTRAG significantly outperforms benchmarks, especially benefiting less advanced LLMs, by providing superior reasoning capabilities and accurate responses.

## 1 Introduction

The emergence of Large Language Models (LLMs) has significantly advanced the field of natural language processing, demonstrating an impressive ability to mimic human-like language patterns [1]. However, despite their extensive knowledge acquired during training, LLMs can occasionally generate factually incorrect information, a phenomenon referred to as "hallucination" [2, 3]. To address this, the integration of retrieval systems with LLMs has been suggested, allowing these models to tap into external databases to generate more reliable responses [4].

Initially, retrieval-augmented generation (RAG) relied on a simple "Retrieve-Read" framework [5], which was adequate for basic question-answering but insufficient for complex, multi-step reasoning tasks. As language models advanced, various prompt-based RAG strategies emerged [6, 7], incorporating pre-retrieval and post-retrieval prompts to refine the process. However, these strategies heavily relied on the foundational capabilities of the language models. Consequently, the focus shifted to Supervised Fine-Tuning (SFT)-based RAG methods [8], which involve fine-tuning language models specifically for RAG tasks to enhance their performance.

While SFT-based methods have improved the quality of generated responses, they face two limitations that hinder their practical application. Firstly, these fine-tuned models are not easily adaptable to

---

\*Equal Contributions;   Correspondence to Zheng Liu and Zhicheng Dou.

38th Conference on Neural Information Processing Systems (NeurIPS 2024).

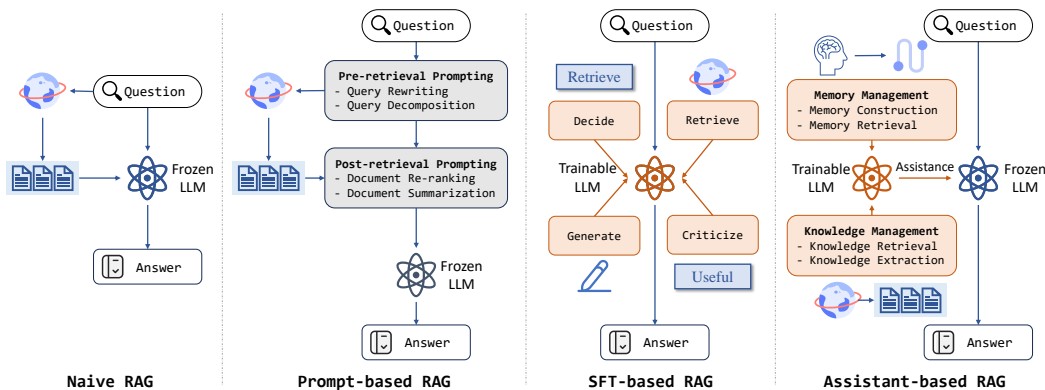

Figure 1: Comparisons of Naive, Prompt-based, SFT-based and our Assistant-based RAG frameworks.

emerging LLMs, requiring retraining for each new foundational LLM. Secondly, directly fine-tuning a foundational LLM in the RAG scenario may change its innate abilities, potentially leading to negative impacts on the model's performance on other tasks. To address these challenges, we propose Assistant-based Retrieval-Augmented Generation (ASSISTRAG), which integrates an intelligent information assistant as a plugin within LLMs. This approach comprises a trainable assistant for information management and a static main LLM dedicated to task execution, as depicted in Figure 1.

As an intelligent information assistant, ASSISTRAG operates in two primary categories to handle complex tasks: memory management and knowledge management. Memory management involves integrating and analyzing content from internal memory, while knowledge management focuses on leveraging external knowledge. These two main functions are supported by four core capabilities of ASSISTRAG: (1) Tool usage, which involves recalling relevant information from both internal memory and external knowledge bases through a retriever; (2) Action execution, which involves processing, analyzing, and extracting information; (3) Memory building, which involves recording essential knowledge and reasoning patterns from historical interactions; (4) Plan specification, which involves determining the necessity of each step in the process. These four capabilities work together to ensure that ASSISTRAG can provide accurate and comprehensive support to the main LLM.

To implement ASSISTRAG, we adopt a two-phase training approach. The first phase, **Curriculum Assistant Learning**, enhances the assistant's capabilities in note-taking, question decomposition, and knowledge extraction through progressively complex tasks. The second phase, **Reinforced Preference Optimization**, uses reinforcement learning to tailor the assistant's feedback to the main LLM's specific needs, optimizing knowledge extraction based on feedback from the main LLM.

During the inference stage, ASSISTRAG operates through a three-step process: (1) Information Retrieval and Integration: The assistant understands the main LLM's needs, retrieves relevant knowledge from internal and external sources, and extracts valuable information. (2) Decision Making: The assistant evaluates and decides whether to provide the retrieved memories and knowledge to the main LLM based on their relevance. (3) Answer Generation and Memory Updating: The main LLM generates an answer using its internal knowledge and the assistant's information, while the assistant updates its memory with crucial reasoning steps.

Results from experiments across three complex question-answering datasets reveal that ASSISTRAG exhibits superior reasoning capabilities and markedly outperforms existing benchmarks. Notably, when applied to different foundational LLMs, ASSISTRAG appears to confer more pronounced benefits on less advanced LLMs.

## 2 Related Work

### 2.1 Retrieval-Augmented Generation

RAG represents a significant advancement in the domain of LLMs, particularly for tasks demanding extensive knowledge. This paradigm begins with a retrieval step, where the LLM accesses an external database to gather relevant information before addressing queries. Traditionally, RAG follows a

"Retrieve-Read" framework [9, 5, 10, 11], with efforts focused on refining either the retriever or the generator through pre-training approaches to augment RAG's accuracy. Building on this foundation, new RAG strategies have emerged, including the use of prompt-based methods like Chain-of-Thought (CoT) reasoning [7, 12], iterative retrieval processes [13, 14, 15, 16], and leveraging LLM-generated content for dynamic retrieval [6, 17, 18]. These strategies underscore the LLMs' ability to select relevant information adaptively in response to specific contexts. Concurrently, research on fine-tuning LLMs for RAG applications is rapidly expanding [19, 20], focusing on enhancing skills such as query reformulation [21] and knowledge integration [22, 23, 24], as well as developing critical functions like determining the necessity of retrieval and appraising the value of retrieved data [8, 25]. Departing from these approaches, our paper introduces Assistant-based RAG, integrating an intelligent information assistant with the main LLM to boost its potential.

## 2.2 LLM-based Autonomous Agents

Recent advancements in LLMs have facilitated the development of LLM-based autonomous agents such as AutoGPT [26], Toolformer [27], and MetaGPT [28], which utilize LLMs for effective decision-making. Notably, ReAct [29] combines LLMs with external tools to manage knowledge-intensive tasks, allowing for dynamic responses to environmental changes. Additionally, models like WebGPT [30] integrate reinforcement learning with GPT-3, enabling the autonomous operation of search engines during text generation. Innovative methods used by Flare [17] and Self-Ask [6] determine optimal times for information retrieval, while Reflexion [31] endows LLMs with introspective mechanisms that continually refine their outputs. Our proposed Assistant-based RAG model further enhances LLM capabilities by combining memory management and knowledge management, thus providing robust support to the main LLM in tackling complex tasks.

## 3 Methodology

In this section, we first define the task of RAG and then introduce our proposed framework, AS-SISTRAG. ASSISTRAG enhances the capabilities of LLMs through the support of an intelligent information assistant. With abilities to use tools, execute actions, build memory, and plan, ASSIS-TRAG can provide precise memory and knowledge management services for LLMs.

### 3.1 Task Definition

Given a question $q$ and a collection of documents $D = \{d_i\}_{i=1}^{|D|}$, the main LLM aims to generate an answer $y$ based on both the question and the relevant documents. This can be formalized as $y = \text{LLM}_{\text{main}}([D_q, q])$, where $D_q$ represents the set of documents retrieved for the query $q$, and $[\cdot, \cdot]$ denotes the concatenation of the retrieved documents with the query. Expanding this concept, ASSISTRAG employs an intelligent information assistant, $\text{LLM}_{\text{Assist}}$, to enhance the main LLM's responses by providing relevant information, formalized as $y = \text{LLM}_{\text{main}}([\text{LLM}_{\text{Assist}}(q), q])$. In the following sections, we will detail the capabilities of the ASSISTRAG framework, along with its training and inference procedures.

### 3.2 ASSISTRAG Overview

By incorporating an intelligent information assistant, ASSISTRAG aims to boost the potential of LLMs in handling complex reasoning tasks. As illustrated in Figure 2, this framework consists of two main components: a frozen main LLM tasked with generating answers based on the information provided, and a trainable assistant LLM responsible for information management. This assistant LLM is designed with two tasks: **Memory Management** involves storing interactions with the main LLM and retrieving relevant past memories to assist in addressing similar questions. **Knowledge Management** encompasses retrieving relevant information from external databases and processing it to support the main LLM in formulating responses to new questions.

To effectively accomplish these tasks, we have endowed the assistant with four key capabilities:

- **Tool Usage:** Retrieving relevant information from internal memory and external knowledge bases.
- **Action Execution:** Reasoning, analyzing information need, and extracting knowledge.
- **Memory Building:** Recording essential knowledge and reasoning patterns from past interactions.

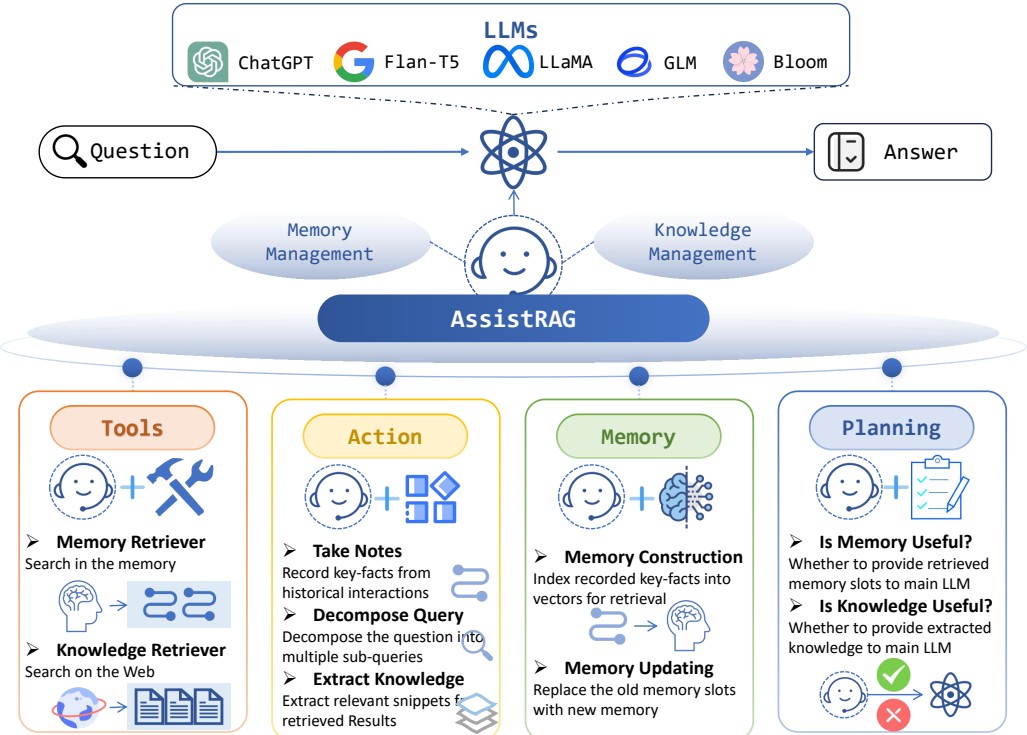

Figure 2: Overview of ASSISTRAG. ASSISTRAG enhances LLMs by providing an intelligent information assistant. Endowed with the ability of tool usage, action execution, memory building and plan specification, it can achieve effective memory and knowledge management.

- **Plan Specification:** Determining the necessity of assistance during answer generation.

These four capabilities synergize to ensure that ASSISTRAG offers precise and comprehensive support to the main LLM. In the following sections, we will provide a detailed examination of the role and implementation of each capability.

### 3.2.1 Memory Management

Effective memory management is crucial for enhancing the main LLM's performance by storing and retrieving historical interactions. This functionality comprises two key processes: capturing new insights and retrieving previously stored information. This stage activates the following three capabilities of AssistRAG:

- **Action I: Note Taking**. This action $\mathcal{F}_{\text{NT}}$ records critical information and the reasoning patterns behind each historical interaction. Given the historical interactions of the main LLM, which include question $q$, reference $r$, and answer $y$, the assistant is tasked with memorizing the key reasoning process behind the answer into the memory slot $m_q$: $m_q \longleftarrow \mathcal{F}_{\text{NT}}(q, r, y)$. The accumulation of memory slots for all prior questions forms the assistant's memory $\mathcal{M}$, which is utilized for subsequent memory retrieval.

- **Tool I: Memory Retriever**. Given the question $q$ and the assistant's memory $\mathcal{M}$, the memory retriever retrieves historically relevant memories, represented as: $\mathcal{M}_q \longleftarrow \mathcal{R}_{\text{memory}}(q, \mathcal{M})$.

- **Plan I: Assessing the Usefulness of Retrieved Memory.** If the question is entirely new, the retrieved memories may not only be unhelpful but also negatively impact the main LLM's response. Therefore, we implement this plan to determine whether the retrieved memory slots should be provided to the main LLM. Using a prompt, the assistant evaluates whether the retrieved memories are beneficial for answering the current question. Only if the answer is affirmative will the retrieved memories be supplied to the main LLM.

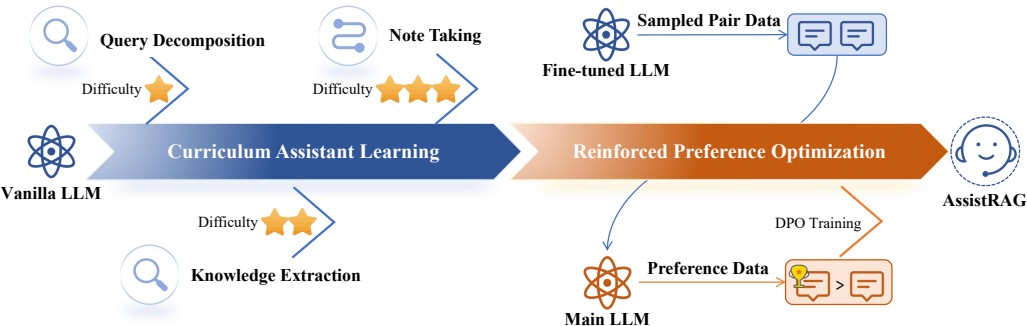

Figure 3: Training framework of ASSISTRAG. It undergoes a two-stage training pipeline through curriculum assistant learning and reinforced preference optimization.

### 3.2.2 Knowledge Management

Effective knowledge management is essential for an intelligent information assistant, involving the efficient gathering of necessary knowledge to support the main LLM. This process includes analyzing the information needs of the main LLM, retrieving relevant knowledge, and integrating it. This process involves the following four capabilities of AssistRAG:

- **Action II: Question Decomposition**. This action $\mathcal{F}_{\text{QD}}$ aims to break down the current question into multiple sub-queries to facilitate the retrieval of knowledge across various aspects: $Q' \longleftarrow \mathcal{F}_{\text{QD}}(q)$, where $Q'$ represents a series of sub-queries derived from the question $q$.

- **Tool II: Knowledge Retriever**. Utilizing a batch of sub-queries $Q'$, the knowledge retriever sources relevant documents from external knowledge bases $D$, denoted as: $D_{Q'} \longleftarrow \mathcal{R}_{\text{knowledge}}(Q', D)$.

- **Action III: Knowledge Extraction**. This action $\mathcal{F}_{\text{KE}}$ involves extracting essential knowledge from a large number of retrieved documents. Given the question $q$ and the retrieved documents $D_{Q'}$, the assistant is responsible for extracting the relevant knowledge $\mathcal{K}_q$ from the search results: $\mathcal{K}_q \longleftarrow \mathcal{F}_{\text{KE}}(q, D_{Q'})$.

- **Plan II: Evaluating the Relevance of Extracted Knowledge.** To ensure the accuracy and relevance of the information provided to the main LLM, this plan determines whether the extracted knowledge should be included in the response generation process. Similarly, we prompt the assistant to assess whether the extracted knowledge is relevant to the current question.

To summarize, we have endowed the assistant with memory capabilities and designed three actions, two retrieval tools, and two planning strategies integral to ASSISTRAG. Next, we will introduce the training strategies developed for ASSISTRAG, focusing on enhancing the accuracy of these actions and ensuring their compatibility with the main LLM.

### 3.3 ASSISTRAG Training

The training objectives of ASSISTRAG focus on two main goals: (1) enhancing the effectiveness of each action within the RAG process, and (2) ensuring that its outputs align with the main LLM's requirements. To achieve these two goals, as depicted in Figure 3, we implement curriculum-based assistant learning and reinforced preference optimization to optimize the training of ASSISTRAG.

Several studies have demonstrated that GPT-4 can achieve human-like annotation accuracy [32]. Based on this consideration, we leverage it to collect training data for the three actions. The supervised training samples for each specific action are cataloged as $\mathcal{C}_{\text{QD}}$, $\mathcal{C}_{\text{KE}}$, and $\mathcal{C}_{\text{NT}}$, preparing these for the assistant's subsequent training phase.

### 3.3.1 Curriculum Assistant Learning

**Motivation.** The tasks of question decomposition, knowledge extraction, and note-taking are interconnected, each contributing towards navigating the reasoning path from a question to its answer.

To equip the assistant with a comprehensive understanding of the RAG process, we devise a step-wise curriculum assistant learning strategy designed to evolve from simpler to more complex tasks to foster a deepened mastery over time.

**Training Objective.** The curriculum learning strategy integrates training samples across three sequential phases $\mathcal{C}_{QD} \rightarrow \mathcal{C}_{KE} \rightarrow \mathcal{C}_{NT}$. Each phase dedicates 60% of its focus to the task at hand, with the remaining 40% evenly divided between the other two tasks. The assistant's training employs the standard next token prediction target based on the training set $D_{gen}$:

$$\mathbb{E}_{(x,y) \sim D_{gen}} \log p_\phi(y|x), \tag{1}$$

where $\phi$ symbolizes the generator's adjustable parameters, and $(x, y)$ is the pair of input and expected output. This methodical training strategy is designed to progressively refine the assistant's proficiency in each component of the RAG process, thereby boosting its effectiveness.

### 3.3.2 Reinforced Preference Optimization

**Motivation.** Although ASSISTRAG effectively handles RAG tasks after assistant learning, its output may sometimes not fully meet the downstream LLM's specific needs. To enhance integration, we implement reinforced preference optimization, a technique that adjusts the assistant's output based on feedback from the main LLM, ensuring tailored assistance that better meets its requirements.

**Training Objective.** To optimize the assistant for better alignment with the main LLM, we adopt Direct Preference Optimization (DPO) [33]. This approach involves generating two sets of references, one from externally retrieved knowledge and the other generated by the assistant itself. The main LLM evaluates these sets, with a preference determined by comparing the F1 scores of its responses against correct answers. For reinforced preference optimization, we leverage the DPO algorithm's optimization objective, utilizing paired preference data $D_{dpo}$:

$$\mathbb{E}_{(x,y_1,y_2) \sim D_{dpo}} \left[\log \sigma \left(\log r_\theta(x, y_1) - \log r_\theta(x, y_2)\right)\right] \tag{2}$$

where $r_\theta(x, y_i) = \beta \frac{\pi_\theta(y_i|x)}{\pi_{ref}(y_i|x)}$ is the reward implicitly defined by the language model $\pi_\theta$ and the reference model $\pi_{ref}$. This reinforced training stage enhances the assistant's capability to deliver assistance that aligns more closely with the main LLM's preferences, enhancing overall efficacy.

### 3.4 ASSISTRAG Inference

Upon completing its training phase, ASSISTRAG initiates its inference process through three steps:

**Information Retrieval and Integration.** At this initial stage, ASSISTRAG first activates Action II to understand the main LLM's information needs. It then uses Tool I and Tool II to retrieve relevant information from internal memory and external knowledge bases, respectively. Subsequently, it invokes Action III to extract essential knowledge from the retrieved documents.

**Decision Making.** In this stage, ASSISTRAG decides whether to provide the retrieved memories and extracted knowledge to the main LLM. It activates Plan I and Plan II to evaluate the relevance and usefulness of the retrieved memories and knowledge for the current question. If the assistant deems them helpful, they are supplied to the main LLM to aid in answer generation.

**Answer Generation and Memory Updating.** In the final phase, we prompt the main LLM to generate an answer based on the question, its internal knowledge, and the information provided by the assistant. Following this, ASSISTRAG activates Action I to utilize its note-taking feature, capturing crucial reasoning steps from the interaction and incorporating them into its memory. This ensures the assistant's knowledge base remains up-to-date.

## 4 Experimental Setup

### 4.1 Datasets and Evaluation Metrics

In this study, we evaluate the effectiveness of our proposed method through experiments on three intricate question-answering datasets: HotpotQA [34], 2WikiMultiHopQA [35], and Bamboogle [6]. These datasets, all derived from Wikipedia documents, provide a uniform corpus and retrieval

Table 1: The evaluation results for three datasets. The "Main LLM" indicates the LLM employed for question answering. The best results are shown in **bold**, while the second-best results are underlined.

| Method | Main LLM | HotpotQA | | | 2Wiki | | | Bamboogle | | |
|---|---|---|---|---|---|---|---|---|---|---|
| | | EM | F1 | Prec. | EM | F1 | Prec. | EM | F1 | Prec. |
| *Baselines without retrieval* | | | | | | | | | | |
| CloseBook | LLaMA2-chat $_{7B}$ | 13.2 | 18.4 | 17.8 | 14.4 | 18.2 | 17.8 | 10.4 | 16.3 | 16.7 |
| CloseBook | ChatGLM $_{6B}$ | 15.6 | 20.4 | 19.9 | 15.8 | 19.5 | 20.0 | 12.6 | 17.6 | 16.9 |
| CloseBook | ChatGPT $_{3.5}$ | 20.0 | 25.8 | 26.4 | 21.6 | 25.7 | 24.5 | 14.4 | 22.0 | 22.3 |
| *Baselines with retrieval* | | | | | | | | | | |
| Naive RAG | LLaMA2-chat $_{7B}$ | 18.2 | 23.0 | 22.5 | 17.4 | 23.7 | 22.8 | 15.2 | 20.4 | 20.3 |
| Naive RAG | ChatGLM $_{6B}$ | 21.8 | 27.2 | 25.8 | 17.8 | 25.0 | 25.2 | 15.8 | 21.1 | 20.8 |
| Naive RAG | ChatGPT $_{3.5}$ | 24.6 | 33.0 | 34.5 | 23.8 | 30.2 | 31.1 | 18.4 | 24.4 | 24.7 |
| ReAct | ChatGPT $_{3.5}$ | 26.8 | 41.7 | 42.6 | 25.0 | 33.0 | 31.6 | 28.8 | 37.7 | 38.2 |
| IRCoT | ChatGPT $_{3.5}$ | 31.4 | 40.3 | 41.6 | 30.8 | 42.6 | 42.3 | 30.2 | 38.8 | 37.9 |
| Self-Ask | ChatGPT $_{3.5}$ | 28.2 | 43.1 | 44.8 | 28.6 | 37.5 | 42.8 | 23.2 | 32.8 | 30.8 |
| SELF-RAG | SELF-RAG $_{7B}$ | 31.0 | 42.4 | 42.3 | 35.0 | 40.7 | 41.0 | 29.8 | 35.5 | 37.8 |
| LLMLingua | ChatGPT $_{3.5}$ | 28.2 | 40.2 | 40.0 | 29.4 | 38.6 | 37.8 | 25.2 | 31.3 | 30.8 |
| ASSISTRAG | LLaMA2-chat $_{7B}$ | 32.4 | 41.5 | 42.6 | 36.2 | 41.0 | 40.5 | 33.0 | 39.6 | 38.7 |
| ASSISTRAG | ChatGLM $_{6B}$ | 33.0 | 42.4 | 43.5 | 38.0 | 43.2 | 42.8 | 32.8 | 39.8 | 39.0 |
| ASSISTRAG | ChatGPT $_{3.5}$ | **34.4** | **44.8** | **46.5** | **39.6** | **45.6** | **45.7** | **34.6** | **41.4** | **41.1** |

mechanisms to supply external references for LLMs. To manage costs, we follow a similar approach as previous studies [17] by selecting a maximum of 500 questions from each dataset's validation set for our experiments. To assess the performance, we employ Exact Match (EM), F1 score, and Precision (Prec.).

### 4.2 Baselines

We benchmark our model against three foundational models: LLaMA2-chat $_{7B}$ [36], ChatGLM3 $_{6B}$ [37], and ChatGPT, assessing their performance in Closebook, Naive RAG, and ASSISTRAG settings. To compare our RAG framework with other RAG models, we include advanced prompt-based methods ReAct [29], IRCoT [7], Self-Ask [6], an SFT-based RAG model Self-RAG [8], and a knowledge extraction model LLMLingua [38]. We ensure a fair comparison by standardizing evaluation conditions across all models.

### 4.3 Implementation Details

**Training Settings:** In the assistant learning phase, we create a dataset comprising 50k training samples based on instruction-following input-output pairs across three distinct task types. The assistant LLM, which is based on ChatGLM3-6B [37], is fully fine-tuned across all parameters. The training is conducted over 2 epochs with a batch size of 32 and a peak learning rate of 2e-5. For the preference optimization phase, we employ a DPO trainer and uses LoRA for fine-tuning, with a learning rate set to 1e-5 and the training duration extended to 2 epochs. The training code and data can be accessed at `https://github.com/smallporridge/AssistRAG`.

**Inference Settings:** For employing ChatGPT, we opt for the `gpt-35-turbo-16k` model, accessed through its API at a temperature setting of 0. We use a Wikipedia dump as our document corpus, breaking down articles into 100-token passages. For both memory and knowledge retrieval, we deploy the off-the-shelf LLM Embedder [39] to fetch up to 5 documents per input.

## 5 Results and Analysis

### 5.1 Main Results

The main results are presented in Table 1. Several key findings can be observed as follows:

**Comparison among Different Reasoning Types.** Applying our ASSISTRAG framework to Chat-GPT demonstrates a significant performance advantage over other models across all datasets. Specifi-

cally, ASSISTRAG showcases its adaptability with different base LLMs, consistently outperforming them in both Closebook and Naive RAG settings. This result highlights the advantage of ASSISTRAG in effectively assisting a variety of downstream LLMs. Additionally, our method surpasses contemporary approaches employing prompt engineering or supervised fine-tuning, validating the efficacy of our curriculum assistant learning and reinforced preference optimization training strategies.

**Comparison among Different Base LLMs.** By comparing the performance of ASSISTRAG across various base LLMs, it is observed that stronger base LLMs yield higher quality responses across all reasoning types. Notably, compared to Naive RAG settings, ASSISTRAG achieves performance improvements of 78%, 51%, and 40% for LLaMA, ChatGLM, and ChatGPT, respectively. This indicates that ASSISTRAG brings more substantial benefits to weaker base LLMs. A likely reason is that weaker models inherently have less robust noise resistance. Benefiting from the assistant's knowledge extraction capability, the main LLM only receives relevant knowledge to generate answers, leading to improved responses.

## 5.2 Analysis

**Ablation Studies.** ASSISTRAG integrates memory and knowledge management to support the main LLM, encompassing three actions: note-taking, question decomposition, and knowledge extraction. To evaluate their contribution, we conduct ablation studies by removing each action or freezing the parameters of the assistant. Additionally, we assess the effects of not implementing planning (w/o. Planning), curriculum learning (w/o. Curriculum), and reinforced preference optimization (w/o. DPO) to explore there contribution to the F1 score. Table 2 illustrates that removing or freezing any of the ASSISTRAG's actions results in decreased performance, underscoring the value of the assistant learning in the RAG context. Notably, maintaining these actions in a frozen state still

Table 2: Ablation Studies of ASSISTRAG.

| Method | Hotpot. | 2Wiki | Bamb. |
|---|---|---|---|
| *Memory Management* | | | |
| Remove $\mathcal{F}_{NT}$ | 40.2 | 42.0 | 39.0 |
| Freeze $\mathcal{F}_{NT}$ | 41.3 | 43.1 | 39.9 |
| *Knowledge Management* | | | |
| Remove $\mathcal{F}_{QD}$ | 39.5 | 37.8 | 37.0 |
| Freeze $\mathcal{F}_{QD}$ | 41.3 | 40.3 | 37.8 |
| Remove $\mathcal{F}_{KE}$ | 39.2 | 38.5 | 38.7 |
| Freeze $\mathcal{F}_{KE}$ | 40.9 | 39.7 | 39.4 |
| ASSISTRAG | 44.8 | 45.6 | 41.4 |
| w/o. Planning | 43.0 | 44.5 | 40.7 |
| w/o. Curriculum | 43.2 | 44.3 | 40.0 |
| w/o. DPO | 42.5 | 43.2 | 40.5 |

outperforms completely removing them, highlighting their critical role in the RAG process. Concerning training strategies, the absence of planning, curriculum learning, and preference optimization slightly diminishes performance, indicating that a structured progression from simple to complex tasks and aligning with downstream LLM preferences contribute to the assistant providing more accurate information to the downstream LLM, thereby enhancing the accuracy of LLM responses.

**Token Usage of Different Methods.** A notable benefit of our ASSISTRAG framework is its efficiency in preprocessing extensive information prior to engaging the main LLM. This not only enhances the inference speed of the main LLM but also minimizes token usage, which is particularly valuable when utilizing online API services like ChatGPT. We select a representative model for each reasoning type to compare their token consumption in terms of online API and SFT model, alongside their performance metrics F1 on the 2Wiki dataset. The results, as outlined in Table 3, reveal significant differences in token

Table 3: Token usage comparison. "tok." is the average token input length preceding the answer.

| Method | API tok. | SFT tok. | F1 |
|---|---|---|---|
| CloseBook | 18 | 0 | 25.7 |
| Naive RAG | 782 | 0 | 30.2 |
| IR-CoT | 1890 | 0 | 42.6 |
| SELF-RAG | 0 | 1456 | 40.7 |
| LLMLingua | 176 | 780 | 38.6 |
| ASSISTRAG | 90 | 1528 | 45.6 |

usage among the methods. Prompt-based RAG methods tend to consume a large number of tokens due to their dependency on multiple API calls. On the other hand, SFT-based methods are more economical in terms of API calls but require retraining for new LLM adaptations. In contrast, our ASSISTRAG demonstrates a balanced approach by reducing API token costs while maintaining adaptability across different LLMs without the need for retraining. This method not only lowers the overall costs associated with API usage but also achieves superior performance.

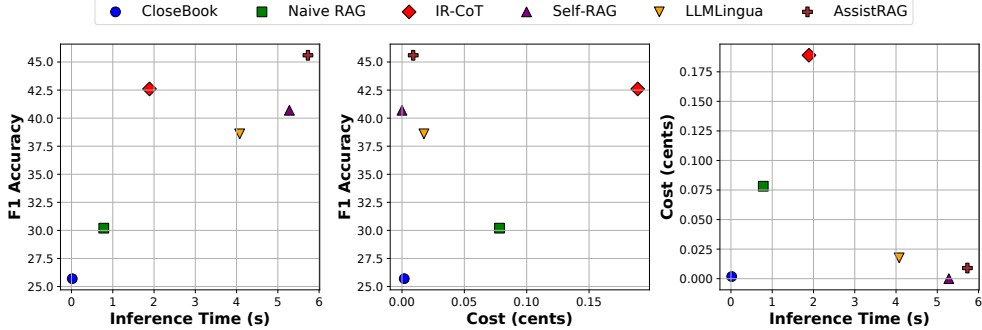

Figure 4: The relationships between inference time, cost, and F1 accuracy for different methods.

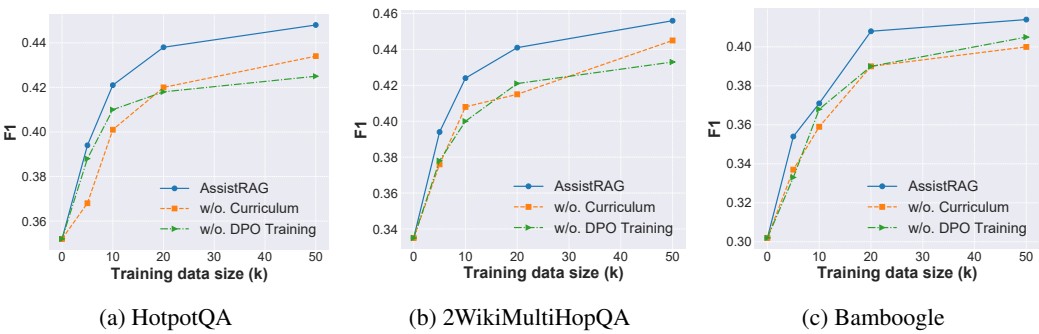

| (a) HotpotQA | (b) 2WikiMultiHopQA | (c) Bamboogle |

Figure 5: Performance with different training data sizes.

**Accuracy, Efficiency, and Cost Analysis.** When evaluating an algorithm's value, we consider three dimensions: accuracy, efficiency, and cost. To compare different RAG methods, we calculate each method's F1 accuracy, inference speed, and cost. We then illustrate the relationships between these variables using three separate plots. From Figure 4, we observe that ASSISTRAG stands out as the most balanced method, achieving the highest F1 accuracy of 45.6, while maintaining a comparable inference time of 5.73 seconds and a low cost of 0.009 cents per question. Although methods like IR-CoT show higher costs and longer inference times, they do not surpass ASSISTRAG in accuracy. These results demonstrate that ASSISTRAG is advantageous for applications requiring high accuracy without incurring significant costs.

**Impact of Dataset Size and Training Strategy.** We examine the effect of training dataset size on model performance by creating subsets of 5k, 10k, and 20k instances from our original 50k training samples. These subsets fine-tune three separate model versions, evaluated on three datasets, and compared to the model trained on the full 50k dataset. We also compare the impact of curriculum learning and DPO training by evaluating performance with each strategy omitted. Figure 5 shows a clear performance improvement for ASSISTRAG as the training dataset size increases from 5k to 50k across both datasets, indicating potential further gains with larger datasets. The curriculum learning strategy performs better than random mixed training, especially with smaller datasets, showing its advantage when data is limited. In contrast, DPO training benefits more from larger datasets, likely because more data enables better training for high-quality data generation.

**Case Study.** Table 4 is a case study that highlights the capabilities of AssistRAG in processing and answering complex comparative questions. In this case study, the main question "Who is older, Danny Green or James Worthy?" is systematically broken down by ASSISTRAG into simpler sub-questions regarding the birth dates of both individuals. This decomposition enables targeted information retrieval, allowing the system to accurately locate and extract relevant birth date information from the corpus. ASSISTRAG effectively retrieves multiple pieces of information, including relevant and irrelevant entries, and filters through them to extract the necessary facts. For instance, it identifies the birth dates of both Danny Green and James Worthy and ignores unrelated entries, such as those concerning another individual named Danny Green who is a boxer. The memory retrieval capability is then utilized to access previous similar questions and their answers, which aids in reinforcing

Table 4: Case Study of AssistRAG.

| |
|---|
| **Question:** Who is older, Danny Green or James Worthy? |
| **Question Decomposition:** AssistRAG initially breaks down the main question into sub-questions to facilitate targeted information retrieval:
- When was Danny Green born?
- When was James Worthy born? |
| **Knowledge Retrieval:** AssistRAG retrieves relevant information from the corpus:
- [1] James Worthy ...
- [2] Danny Green (basketball) ...
- [3] Danny Green (boxer) ...
- ... |
| **Knowledge Extraction:** From the retrieved information, AssistRAG extracts key facts:
- Danny Green (born June 22, 1987) is an American professional basketball player who last played for the Philadelphia 76ers of the National Basketball Association (NBA). In his NBA career, Green has played for six teams. As of 2020, Green is one of just four players in history to have won NBA championships with three different teams.
- James Worthy James Ager Worthy (born February 27, 1961) is an American former professional basketball player who is currently a commentator, television host, and analyst. A standout at the University of North Carolina, the 6 ft 9 in (2.06 m) small forward shared College Player of the Year honors en route to leading the Tar Heels to the 1982 NCAA Championship. |
| **Memory Retrieval:** AssistRAG retrieves similar questions and answers from the memory:
1. question: Who is older, Danny Jones or David Coverdale? answer: David Coverdale was born on 22 September 1951, while Danny Jones was born on 12 March 1986. Since David Coverdale was born earlier, he is older than Danny Jones. So the answer is David Coverdale.
2. question: Who is older, Danny Shirley or Kevin Parker? answer: Danny Shirley was born on August 12, 1956, while Kevin Parker was born on January 20, 1986. Since Danny Shirley was born earlier than Kevin Parker, he is older. So the answer is Danny Shirley.
3. ... |
| **Planning:** Both retrieved memory and extracted knowledge are useful for the question. |
| **Final Output:** Based on the analysis, Main LLM outputs:
- James Worthy was born on February 27, 1961, while Danny Green was born on June 27, 1987. Since James Worthy was born earlier, he is older. So the answer is James Worthy. |
| **Final Answer:** James Worthy |
| **Ground-truth:** James Worthy |

the current decision-making process. This step provides context and supports consistency in the reasoning pattern applied by the system. Combining the extracted knowledge and retrieved memory, ASSISTRAG plans the final response by confirming the usefulness of both sources of information. The Main LLM then generates a comprehensive answer, stating that James Worthy, born on February 27, 1961, is older than Danny Green, born on June 22, 1987. This case study showcases ASSISTRAG's ability to manage complex tasks by leveraging its multi-step reasoning process, ensuring the delivery of accurate and reliable answers. The superior performance in accurately answering comparative questions is a testament to the system's robust architecture and its effective integration of question decomposition, information retrieval, knowledge extraction, planning, and memory capabilities.

# 6 Conclusion

In this study, we introduce ASSISTRAG to augment LLMs with an intelligent information assistant, significantly improving their ability to tackle tasks requiring complex reasoning. By implementing a two-stage training methodology that integrates curriculum assistant learning with reinforced preference optimization, we enhance the assistant's skills in memory and knowledge management. Experiments demonstrate that ASSISTRAG surpasses existing baselines with a notable margin. Looking ahead, we plan to further expand the assistant's skills to include long-text processing [40] and personalized support [41], thereby providing more effective assistance to the main LLM.

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

## A    Appendix / supplemental material

### A.1    Implementation Details

To achieve the effective functioning of ASSISTRAG, we meticulously designed and executed the training and inference phases, ensuring optimal use of computational resources and robust fine-tuning processes.

### A.1.1    Training Settings

The training of the assistant LLM involved a two-phase approach: Assistant Learning and Preference Optimization.

1. **Assistant Learning Phase:**

- **Dataset Creation:** We created a dataset comprising 50,000 training samples based on instruction-following input-output pairs. These pairs were categorized across three distinct task types: question decomposition, note-taking, and knowledge extraction.
- **Model Architecture:** The assistant LLM is based on ChatGLM3-6B [37], a state-of-the-art language model known for its robust performance in various NLP tasks.
- **Training Procedure:** The assistant LLM was fully fine-tuned across all parameters over 2 epochs, with a batch size of 32 and a peak learning rate of 2e-5. This phase focused on enhancing the model's ability to decompose complex queries, take notes, and extract relevant knowledge.

Table 5: Training samples of note-taking.

**Input**

You are given 1) the question, 2) the answer, 3) the supporting facts where the answer can be derived. You are supposed to figure out the reasoning process towards the answer step-by-step without other content. Be concise and direct.

**Question:** American politician Joe Heck ran unsuccessfully against Democrat Catherine Cortez Masto, a woman who previously served as the 32nd Attorney General of where?

**Answer:** Nevada

**Supporting Facts:** Joseph John Heck (born October 30, 1961) is an American politician, physician, and U.S. Army Brigadier General who had served as the U.S. Representative for Nevada's 3rd congressional district from 2011 to 2017. He ran unsuccessfully against Democrat Catherine Cortez Masto in the general election for the open Nevada United States Senate seat in 2016. She previously served as the 32nd Attorney General of Nevada from 2007 to 2015.

**Reasoning:**

**Output**

Catherine Cortez Masto previously served as the 32nd Attorney General of Nevada. Joe Heck ran unsuccessfully against her in the general election for the open Nevada United States Senate seat in 2016. Therefore, the answer is Nevada."

2. **Preference Optimization Phase:**

- **Optimization Technique:** We employed a DPO (Distributed Preference Optimization) trainer to refine the assistant's feedback mechanisms.
- **Fine-tuning with LoRA:** Low-Rank Adaptation (LoRA) was utilized for fine-tuning, which helps in adjusting a subset of model parameters efficiently, reducing the computational load.
- **Learning Rate and Duration:** The learning rate was set to 1e-5, with the training duration extending to 2 epochs.

**Training Resources:** The entire training process was conducted using 8 A800 GPUs, providing substantial computational power to handle the intensive training tasks efficiently.

### A.1.2   Inference Settings

During the inference phase, the ASSISTRAG system was fine-tuned to operate seamlessly with the main LLM, ensuring effective retrieval and processing of information.

- **Model Selection:** For the main LLM, we opted for the `gpt-35-turbo-16k` model, accessed through its API. This model was chosen for its extended context window and advanced capabilities in handling complex queries.
- **Temperature Setting:** The temperature was set to 0, ensuring deterministic outputs which are crucial for consistency in inference tasks.
- **Document Corpus:** A Wikipedia dump was used as the primary document corpus. Articles were segmented into 100-token passages to facilitate efficient retrieval.
- **Retrieval tool:** Both memory and knowledge retrieval processes utilized the off-the-shelf LLM Embedder [39], capable of fetching up to 5 documents per input, ensuring comprehensive coverage of relevant information.

**Inference Resources:** The inference tasks were efficiently handled using a single A800 GPU, highlighting the system's capability to deliver high performance with minimal resource usage.

### A.1.3   Data Annotation and Samples

The training data annotation process was critical in ensuring the quality and relevance of the dataset. We utilized GPT4-turbo for annotating the dataset, ensuring high accuracy and consistency across the training samples. The dataset included approximately 50k training samples, labeled to cover a wide range of instruction-following tasks.

**Instructions and Training Data Samples:** The following tables (Table 6, 5, and 7) provide detailed examples of the instruction types and training data samples used in the training phase.

Table 6: Training samples of query decomposition.

| |
|---|
| **Input** |
| Please generate a series of search queries that can be used to find information relevant to the given question. **Question:** The 1968 Baltimore Orioles season included a replacement of Hank Bauer by the manager who spent how many seasons with the Orioles?. **Search queries:** |
| **Output** Who replaced Hank Bauer as the manager of the 1968 Baltimore Orioles? How many seasons did Earl Weaver spend with the Baltimore Orioles? |

Table 7: Training samples of knowledge extraction.

| |
|---|
| **Input** |
| Please extract relevant snippets from search results that would be helpful in answering this question. **Question:** Musician and satirist Allie Goertz wrote a song about the The Simpsons character Milhouse, who Matt Groening named after who?. **Search results:** - [1] Allie Goertz – Allie Goertz Allison Beth Allie Goertz (born March 2, 1991) is an American musician. Goertz is known for her satirical songs based on various pop culture topics. Her videos are posted on YouTube under the name of Cossbysweater. Subjects of her songs have included the film The Room ... - [2] Allie Goertz – and logo of the series. The album was made possible through Kickstarter. She is co-host of Everything's Coming Up Simpsons podcast, formerly known as Everything's Coming Up Podcast, a Simpsons-focused podcast along with Julia Prescott. Goertz also served as a social media producer for ... - [3] ... **Snippets:** |
| **Output** Allie Goertz – Allie Goertz Allison Beth Allie Goertz (born March 2, 1991) is an American musician. Goertz is known for her satirical songs based on various pop culture topics. Milhouse Van Houten Milhouse Mussolini Van Houten is a fictional character in the animated television series The Simpsons, voiced by Pamela Hayden, and created by Matt Groening who named the character after ... |

## A.2 Error Analysis

We have conducted a comprehensive error analysis on the performance of AssistRAG. To facilitate this analysis, we selected 50 erroneous examples from the HotpotQA dataset and calculated the proportion of each error type:

1. **Insufficient Knowledge Retrieval**: Instances where the retrieved knowledge does not contain the answer.

2. **Knowledge Extraction Errors**: Cases where the answer is present in the retrieved knowledge, but the assistant fails to extract this information.

3. **Answer Reasoning Mistakes**: Situations where the assistant extracts the correct information but the main LLM produces an incorrect answer.

4. **Other**: Including errors such as non-exact match answers.

From Table 8, our findings indicate that more than half of the errors stem from insufficient knowledge retrieval, which is likely linked to the performance of the retriever and the manner in which questions are reformulated. Additionally, a significant portion of errors are due to reasoning mistakes, highlighting the importance of the main LLM's reasoning capabilities. Given that HotpotQA involves multi-hop question-answering tasks, these findings underscore the high demands placed on reasoning abilities.

## A.3 Limitations

Despite the advancements offered by ASSISTRAG, several limitations warrant consideration. Firstly, relying on an intelligent information assistant introduces additional computational complexity and latency. The two-phase training approach and its operation during inference require substantial computational resources, which may limit the practical application of ASSISTRAG in environments with restricted processing capabilities or where real-time responses are critical.

Table 8: Proportion of each error type in the analysis.

| Error Type | Proportion |
|---|---|
| Insufficient Knowledge Retrieval | 58% |
| Knowledge Extraction Errors | 12% |
| Answer Reasoning Mistakes | 20% |
| Other | 10% |

Secondly, the effectiveness of ASSISTRAG depends on the quality and comprehensiveness of the external knowledge bases and internal memory it accesses. In scenarios where the available data is sparse, outdated, or biased, the assistant's ability to retrieve and integrate relevant information may be compromised, leading to suboptimal or erroneous outputs from the main LLM. This dependence on data quality underscores the need for continuous updates and maintenance of the knowledge sources.

Lastly, the decision-making process during inference, which involves the assistant evaluating the relevance of retrieved information, is inherently difficult. The assistant's ability to accurately determine the necessity and applicability of specific knowledge is crucial for effective support. However, this process is susceptible to errors, particularly in scenarios involving ambiguous or multifaceted queries. Enhancing the precision and reliability of this decision-making mechanism is a key area for further research.

