# OpenReview forum: "Boosting the Potential of Large Language Models with an Intelligent Information Assistant"
_NeurIPS.cc/2024/Conference — NeurIPS 2024 poster_

### Official Review · Reviewer_zFfj · 2024-07-08

**Soundness:** 3
**Presentation:** 2
**Contribution:** 2
**Rating:** 5
**Confidence:** 4

**Summary:**

The paper introduces AssistRAG, a framework that integrates an intelligent information assistant within LLMs. AssistRAG employs a two-phase training approach involving Curriculum Assistant Learning and Reinforced Preference Optimization, focusing on memory management and knowledge management. Experiments on three QA datasets demonstrate the effectiveness.

**Strengths:**

1. Experiments show the proposed method is effective.

**Weaknesses:**

1. Questionable innovation, the key ideas of the proposed method (memory and knowledge management) become a standard practice in terms of Agent design.
2. The proposed method involves model training and serving, which could be costly, and also make it harder to adapt to different foundation LLMs.
3. A.2 shows that 58% of the errors stem from insufficient knowledge retrieval, but no further details are provided. Given such a scenario, the retrieval strategy may need to be improved.
4. Using GPT-4 to generate fine-tuned data potentially limits the proposed method.
5. Notations are unclear, e.g., line 97, what is d_i? Eq. 2 is not well explained.
6. Table 1 shows that AssistRAG is combined with LLaMA2, ChatGLM, and ChatGPT, but the manuscript only mentioned using ChatGLM3 for training and ChatGPT in inference, which is inconsistent.

**Questions:**

1. Please include more details about the (ChatGPT) model versions employed in this paper.
2. The articles were segmented into 100-token passages - this is a relatively short length for LLM, and A.2 also shows that insufficient knowledge retrieval caused more than half of the errors, why don't consider increasing the chunk size?
3. Experiment data and questions are focused on Wikipedia, how about other domains?

**Limitations:**

1. Given the current development of the Agent framework, it is necessary to compare the proposed method with the Agent solution.

---

> ### Author Rebuttal · Authors · 2024-08-06
>
> Dear Reviewer,
>
> Thank you for your valuable feedback. We have responded to each of the weaknesses (W), questions (Q), and limitations (L) you raised. We hope the following responses clarify the contributions of our work and address your concerns.
>
> ***
> **R to W1:**
>
> While memory and knowledge management are common elements in agent design, our approach introduces significant innovations that set it apart:
>
> 1. AssistRAG is the first to integrate a complete agent solution within RAG scenarios, encompassing tool usage, action execution, memory building, and plan specification. This sets a foundational approach for future intelligent assistants for human.
>
> 2. By decoupling the RAG task into an Assistant LLM and a Main LLM, AssistRAG enhances the assistant's adaptability to RAG scenarios without compromising the main LLM's inherent capabilities through a two-phase training process.
>
> 3. AssistRAG adapts efficiently to new LLMs without the need for retraining from scratch, achieving significant improvements with preference alignment.
>
> The following table highlights the distinctions between AssistRAG and other representative works:
>
> |Model|Tool|Action|Memory|Plan|Training for RAG|No Impact on Main LLM|Adaptation to new LLMs|
> |-|:-:|:-:|:-:|:-:|:-:|:-:|:-:|
> |ReAct|√|√|×|√|×|√|No training|
> |Self-RAG|√|√|×|√|√|×|Training from scratch|
> |Selfmem|√|×|√|×|√|×|Training from scratch|
> |AssistRAG|√|√|√|√|√|√|Preference alignment|
>
> ***
> **R to W2:**
>
> We understand your concerns regarding the cost and adaptability of the proposed method. In fact, our Assistant-based RAG framework is specifically designed to improve adaptability to different foundation LLMs.
>
> 1. Decoupling the RAG task into an Assistant LLM and a Main LLM is intended to separate the training of the Assistant LLM from the Main LLM. The Assistant Learning phase remains unchanged regardless of the Main LLM. This allows a well-trained Assistant LLM to be adapted to various Main LLMs without the need for retraining from scratch.
> 2. The DPO training phase is designed to enhance compatibility between the Assistant and Main LLMs. Even without this phase, the Assistant LLM can still effectively support the Main LLM, as demonstrated in Table 2 (No DPO Training).
>
> ***
> **R to W3:**
>
> There may have been a misunderstanding regarding our presentation. We hope the following points clarify this:
>
> 1. The 58% refers to the proportion of errors caused by insufficient knowledge retrieval among all incorrect answers, not all questions.
> 2. We employ a highly advanced and widely used model LLM-Embedder for retrieval, which is proposed in 2023 and has been downloaded over 56k times on Huggingface.
> 3. Our focus is on complex multi-hop QA tasks, which require retrieving at least two correct documents to ensure knowledge completeness. This increases the difficulty for the retriever.
>
> ***
> **R to W4:**
>
> We would like to clarify the following points to address your concerns regarding the use of GPT-4:
>
> 1. Using GPT-4 to generate fine-tuned data has been widely adopted in various scenarios, such as WizardLM, WizardCoder, WizardMath.
> 2. Our proposed method does not rely exclusively on GPT-4. Recently released powerful open-source models, such as Llama-3.1 and DeepSeek-V2, can serve as alternatives.
> 3. We will open-source our training data and models. Community researchers will be able to adapt our method to their own LLMs without needing to use GPT-4 for generating training data.
>
> ***
> **R to W5:**
>
> We apologize for the unclear notation. We will revise the manuscript to provide clearer definitions and explanations for the notations.
>
> ***
> **R to W6:**
>
> We acknowledge the inconsistency you pointed out between Table 1 and section 4.3 (Inference Settings). To clarify, the base model used for training the Assistant LLM is ChatGLM3. In order to verify that the Assistant LLM can adapt to various main LLMs, we conducted inference using LLaMA2, ChatGLM, and ChatGPT, which also involved providing preference data. We will improve our wording in future versions to prevent any misunderstandings.
>
> ***
> **R to Q1:**
>
> We appreciate your request for additional details regarding the model versions used. The specific ChatGPT model employed in this paper is gpt-35-turbo-16k, provided by Azure, with a release date of 2023-05-15.
>
> ***
> **R to Q2:**
>
> There are several reasons for choosing 100-token wikipedia passages as the retrieval corpus:
>
> 1. Most baselines in its original papers used this setting, allowing for easier comparison and alignment with these baselines.
> 2. Wikipedia's official passage-level retrieval corpus segments articles into 100-token passages.
> 3. We do not input just one passage into the LLM but combine multiple retrieved passages. This length ensures that the combined input does not exceed the length limits of certain LLMs.
>
> ***
> **R to Q3:**
>
> This is indeed a valuable suggestion. We have expanded our study to include the ALCE-ELI5 dataset, which originates from the Reddit forum. This dataset is built upon the Sphere corpus, encompassing 899 million passages. The results are as follows:
>
> |Main LLM|Method|Fluency|Correctness|
> |-|-|-|-|
> |LLaMA-2-chat|CloseBook|50.3|20.3|
> |LLaMA-2-chat|Naive RAG|66.8|27.1|
> |LLaMA-2-chat|AssistRAG|65.9|33.5|
> |ChatGPT|LLMLingua|67.2|41.4|
> |ChatGPT|CloseBook|53.3|37.2|
> |ChatGPT|Naive RAG|67.4|40.0|
> |ChatGPT|AssistRAG|**67.8**|**45.4**|
>
> ***
> **R to L1:**
>
> We have conducted additional experiments comparing our proposed method with two Agent frameworks capable of performing RAG tasks: Toolformer and Reflexion. These experiments were conducted on the 2wiki dataset, and we will supplement results from other datasets in later version.
>
> | |EM|F1|Recall|
> |-|-|-|-|
> |Toolformer|27.2|38.6|43.2|
> |Reflexion|31.8|41.7|44.2|
> |AssistRAG|**39.6**|**45.6**|**45.7**|
>
> ***
> We hope these clarifications address your concerns and provide a better understanding of our work. If you have any further concerns, we would be delighted to continue the discussion with you.

---

> > ### Comment · Reviewer_zFfj · 2024-08-09
> > **Response to authors**
> >
> > Thank you for your response and the additional information provided. After carefully considering the paper and the rebuttal content, I have decided to slightly increase my evaluation, based on a comprehensive assessment of the work's contribution to the field.
> >
> > I would like to further clarify my concern regarding the insufficient knowledge retrieval. The current experimental results indicate that insufficient knowledge retrieval accounts for 58% of the errors, suggesting potential issues at the retrieval stage. This problem typically does not stem from the retrieval algorithm, package, or vector database, but rather from the data processing workflow. This raises questions about the use of 100-token passages, as current best practices typically involve chunk sizes in the range of 500-1000 tokens to capture more semantic content.
> >
> > Thank you for your efforts, and good luck.

---

> > > ### Author Response · Authors · 2024-08-11
> > >
> > > Thank you for your thorough evaluation and for considering the content of our rebuttal. We sincerely appreciate your willingness to reevaluate our work and the constructive feedback you’ve provided.
> > > ***
> > > Regarding your concern about insufficient knowledge retrieval, this is indeed an interesting issue. We conducted experiments by increasing the chunk size to 512 tokens while keeping the number of retrieved documents constant. The results show the following changes in error proportions:
> > >
> > > | Error Type | Original Proportion | New Proportion |
> > > |------------|---------------------|----------------|
> > > | Insufficient Knowledge Retrieval | 58% | 48% |
> > > | Knowledge Extraction Errors      | 12% | 20% |
> > > | Answer Reasoning Mistakes        | 20% | 22% |
> > > | Other                             | 10% | 10% |
> > >
> > > Increasing the chunk size did reduce the proportion of insufficient knowledge retrieval errors but also led to a higher likelihood of knowledge extraction errors due to the introduction of more irrelevant context. This trade-off is worth further exploration. Our opinion is that for models with strong knowledge extraction capabilities, increasing the chunk size or the number of retrieved documents can be an effective strategy.
> > > ***
> > > Once again, thank you for your invaluable insights and your support of our work. Your recognition is our greatest encouragement.

---

### Official Review · Reviewer_Y6YL · 2024-07-13

**Soundness:** 3
**Presentation:** 3
**Contribution:** 3
**Rating:** 7
**Confidence:** 3

**Summary:**

To address the limitation of LLMs generating factually incorrect information, the authors have introduced AssistRAG, an intelligent information assistant with LLMs, building upon existing retrieval-augmented generation (RAG) strategies. The system operates in two main categories: memory management and knowledge management. AssistRAG employs a two-phase training approach: Curriculum Assistant Learning and Reinforced Preference Optimization. During inference, AssistRAG follows a three-step process: information retrieval and integration, decision-making, and answer generation with memory updating.

**Strengths:**

1. Novel integration of curriculum assistant learning and reinforced preference optimization, distinguishing it from traditional approaches.
2. AssistRAG demonstrates consistent outperformance across multiple datasets and base models, highlighting its potential for broad applications in improving LLM capabilities.

**Weaknesses:**

1. The paper might benefit from addressing potential scalability issues and providing more extensive comparisons with a broader range of state-of-the-art models.
2. Including real-world application scenarios and discussing potential limitations or challenges in deploying AssistRAG in practical settings would also enhance the paper's impact and applicability.

**Questions:**

1. How does AssistRAG perform in terms of computational efficiency and scalability when applied to very large datasets or in real-time applications? Can the authors provide any benchmarks or comparisons?
2. The ablation studies are informative, but could the authors include additional analysis on the sensitivity of AssistRAG to different hyperparameters or training configurations?
3. Incorporate user studies or qualitative evaluations to assess the practical usability and effectiveness of the assistant in aiding human users.

**Limitations:**

1. Llama 2 Chat (7B parameters) and ChatGLM3 (6B parameters) have substantially fewer parameters than GPT-3.5, which is reported to have a much higher parameter count. Moreover, GPT4-turbo is used for annotating the dataset (as mentioned in Appendix section A.1.3), however, the model isn’t used, or mentioned, in the experiments. This disparity in model size should be taken into account when comparing the performance of these models, particularly in relation to GPT-3.5.
2. The analysis done on token usage (section 5.2) is only conducted on the 2WikiMultiHopQA dataset, however other experiments in the study are done on the HotpotQA, 2WikiMultiHopQA, and Bamboogle datasets. The paper could benefit from a consistent dataset used for all evaluations.

---

> ### Author Rebuttal · Authors · 2024-08-06
>
> Thank you very much for your valuable feedback and recognition of our paper. We hope that the following responses will address your concerns:
>
> ***
> **R to W1:**
>
> To further demonstrate the effectiveness of our model, we have included additional experiments with two agent solutions on the 2wiki dataset. The results are as follows:
>
> |            | EM       | F1       | Recall   |
> | ---------- | -------- | -------- | -------- |
> | Toolformer | 27.2     | 38.6     | 43.2     |
> | Reflexion  | 31.8     | 41.7     | 44.2     |
> | AssistRAG  | **39.6** | **45.6** | **45.7** |
>
> These results show that AssistRAG outperforms both Toolformer and Reflexion, demonstrating its superior performance in RAG tasks.
>
> ***
> **R to W2:**
>
> Thank you for your insightful suggestion. In Appendix A.4, we have discussed some potential limitations and challenges in deploying AssistRAG in practical settings. We will provide a more detailed discussion in future versions to further enhance the paper's impact and applicability following your advice.
>
> ***
> **R to Q1:**
>
> Computational efficiency in real-time applications is indeed crucial. To address the concerns, we first analyze the composition of inference time when encountering a new question:
>
> 1. The time taken by the Assistant LLM to perform actions, including Question Decomposition and Knowledge Extraction.
> 2. The time taken by the Assistant LLM to call retrievers, including memory retrieval and knowledge retrieval.
> 3. The time taken by the Main LLM to generate the answer.
>
> Among these, steps 1 and 3 are independent of dataset size. Dataset size only affects step 2. In step 2, to achieve low-latency retrieval, we have utilized the FAISS library to index the documents and employed an IVF index structure for acceleration. This allows for millisecond-level retrieval speed even with datasets in the millions of documents range. Compared to the seconds-level time taken by steps 1 and 3, the retrieval time in step 2 is negligible. Therefore, very large datasets (below the billion-level) would not significantly impact computational efficiency.
>
> ***
> **R to Q2:**
>
> Thank you for your insightful suggestion. We have conducted additional experiments to analyze the sensitivity of AssistRAG (ChatGLM 6B) to different hyperparameters, specifically the learning rate during the Assistant Learning phase and the number of documents retrieved (K) during the knowledge retrieval phase. The results are as follows:
>
> | Top K     | K=1  | K=2  | K=3  | K=4  | K=5  |
> | --------- | ---- | ---- | ---- | ---- | ---- |
> | F1| 20.4 | 36.6 | 39.8 | 42.4 | 43.2 |
>
> | Learning Rate | lr=2e-4 | lr=1e-4 | lr=5e-5 | lr=2e-5 | lr=1e-5 |
> | ------------- | ------- | ------- | ------- | ------- | ------- |
> | F1     | 40.6    | 41.4    | 42.0    | 43.2    | 42.6    |
>
> We hope these additional analyses provide a clearer understanding of how AssistRAG performs under different hyperparameter settings and training configurations.
>
> ***
> **R to Q3:**
>
> This is indeed a valuable suggestion. Following your recommendations, we conducted a user study involving three participants. Each participant was asked to write 20 factual questions. Subsequently, answers were generated using ChatGPT in three configurations: CloseBook, Naive RAG, and AssistRAG. The participants then evaluated their satisfaction with the answers provided. The results are as follows:
>
> |           | User A   | User B   | User C   |
> | --------- | -------- | -------- | -------- |
> | CloseBook | 0.25     | 0.20     | 0.30     |
> | Naive RAG | 0.45     | 0.55     | 0.45     |
> | AssistRAG | **0.70** | **0.75** | **0.65** |
>
> These results demonstrate that users were more satisfied with the answers generated by AssistRAG compared to the other configurations. We believe this user study highlights the practical usability and effectiveness of AssistRAG in aiding human users.
>
> ***
> **R to L1:**
>
> We apologize for the confusion. Our intention in selecting models with varying parameter counts was to demonstrate that AssistRAG significantly outperforms CloseBook and Naive RAG inference methods, regardless of the model size. We appreciate your suggestion and agree that presenting the results by grouping models of similar sizes will make the comparisons clearer. We will revise Table 1 to reflect this improvement, ensuring that models with similar parameter counts are compared directly.
>
> ***
> **R to L2:**
>
> Thank you for your suggestion. In fact, we have conducted token usage analysis across all three datasets. Since these datasets share the same retrieval corpus, the lengths of the retrieved passages are similar, resulting in no significant differences in token usage across different datasets. Therefore, we chose to present the representative 2Wiki results. For consistency, we will include token usage results for all three datasets in future versions of the paper.
>
> ***
> We greatly appreciate the time and effort you have invested in reviewing our manuscript. Your insightful comments have been invaluable in helping us improve the quality and clarity of our paper. Thank you once again for your constructive feedback.

---

### Official Review · Reviewer_xG33 · 2024-07-19

**Soundness:** 4
**Presentation:** 4
**Contribution:** 4
**Rating:** 8
**Confidence:** 4

**Summary:**

This paper proposes AssistRAG, an architecture for augmenting LLMs with a separate, trainable agent that helps with information retrieval and memory/knowledge management. The authors motivate such an architecture (as opposed to, say, fine-tuning the main LLM for RAG) and describe how to build and train it. The conduct experiments with 3 information heavy benchmarks and show consistent performance gains compared to a number of recent baselines.

**Strengths:**

* The motivation behind AssistRAG is strong -- directly fine-tuning the main LLM for retrieval augmented generation is costly and can/does negatively impact its other abilities. Thus, freezing the main LLM and instead training a smaller auxiliary LLM to decide what to retrieve (tailored to the needs of the main LLM) and how to manage retrieved information is a highly desirable approach.

* The paper is very clearly written. The structure is easy to follow and balanced. The motivation, as noted above, is well articulated. The baselines and method are clearly described and the results well-summarized.

* Ablations show the value of individual components, and additional experiments confirm the benefits of the proposed approach even as the amount of training data is varied.

**Weaknesses:**

I don't see any notable weaknesses in this paper. It's possible that I am not aware of the latest in this line of work and my assessment of novelty may not be accurate. I will defer to the other reviewers on the novelty aspect.

Some of the datasets used in this paper are rather old -- they are from the era of the train-test paradigm with 10s of thousands of training instances (e.g., HotpotQA was published in 2018!). I realize they are being used here in the model context of prompted LLMs. Nevertheless, I wonder if there are newer datasets that might be more timely.

**Questions:**

Please see the weakness section.

**Limitations:**

Yes

---

> ### Author Rebuttal · Authors · 2024-08-06
>
> Thank you for your recognition of our paper. We sincerely appreciate all the feedback from the reviewers and will make revisions to enhance the paper's impact and applicability based on your valuable suggestions.
>
> Regarding the datasets, we have compiled the publication years of all commonly used QA datasets, as shown in the table below:
>
> | Year | Datasets                                                |
> | ---- | ------------------------------------------------------- |
> | 2023 | Bamboogle                                               |
> | 2022 | PopQA, ASQA, Musique                                    |
> | 2020 | 2WikiMultiHopQA                                         |
> | 2019 | SIQA, NQ, CommenseQA, BoolQ, PIQA, Fermi, ELI5, AmbigQA |
> | 2018 | HotpotQA, NarrativeQA, WikiQA                           |
> | 2017 | TriviaQA, MSMARCO-QA                                    |
> | 2016 | SQuAD                                                   |
>
> As our paper focuses on complex QA tasks, we selected Multihop QA datasets for our study. These include Bamboogle (2023), 2WikiMultiHopQA (2020), and HotpotQA (2018). Among these, Bamboogle is the most recent Multihop QA dataset available.
>
> Thank you once again for your constructive feedback. We greatly appreciate the time and effort you have invested in reviewing our manuscript.

---

> > ### Comment · Reviewer_xG33 · 2024-08-13
> > **Re: Rebuttal by Authors**
> >
> > Thank you for the response. Given your focus on Multihop QA datasets, `Musique` might have been a good one to try as it has some harder questions requiring 3 or 4 hops. Nevertheless, after looking at the other reviews and your responses, I remain in favor of accepting this paper.  The only other suggestion I have is to include a clearer comparison with other similar work in the updated version of the paper.  Thank you for the work!

---

> > > ### Author Response · Authors · 2024-08-14
> > >
> > > Thank you for your positive feedback and valuable suggestions. We will explore integrating the Musique dataset and expand our comparison part to provide a clearer analysis of similar work in the updated version. We appreciate your support and look forward to enhancing our paper based on your recommendations.

---

### Decision · Program_Chairs · 2024-09-25

**Decision:**

Accept (poster)

**Comment:**

The reviewers agree that this paper addresses an important problem and makes a significant contribution to the existing literature on RAG-based systems. No major weaknesses were identified.

If ultimately accepted, I expect the authors to integrate the reviewer feedback into the final version.